# DYNAMICALLY ADAPTING FLOATING-POINT PRECISION TO ACCELERATE DEEP NEURAL NETWORK TRAINING

## ABSTRACT

Mixed-precision arithmetic combining both single- and half-precision operands on the same operation has been successfully applied to train deep neural networks. Despite the advantages of mixed-precision arithmetic in terms of reducing the need for key resources like memory bandwidth or register file size, it has a limited capacity for diminishing computing costs and requires 32-bits to represent its output operands. This paper proposes two approaches to replace mixed-precision for half-precision arithmetic during a large portion of the training. The first approach achieves accuracy ratios slightly slower than the state-of-the-art by using half-precision arithmetic during more than 99% of training. The second approach reaches the same accuracy as the state-of-the-art by dynamically switching between half- and mixed-precision arithmetic during training. It uses half-precision during more than 94% of the training process. This paper is the first in demonstrating that half-precision can be used for a very large portion of DNNs training and still reach state-of-the-art accuracy.

## 1 INTRODUCTION

The use of Deep Neural Networks (DNNs) is becoming ubiquitous in areas like computer vision (Krizhevsky et al., 2012; Szegedy et al., 2015), speech recognition (Hinton et al., 2012), or language translation (Wu et al., 2016). DNNs display very remarkable pattern detection capacities and, more specifically, Convolutional Neural Networks (CNNs) are able to accurately detect and classify objects over very large image sets (Krizhevsky et al., 2012). Despite this success, a large amount of samples must be exposed to the model for tens or even hundreds of times during training until an acceptable accuracy threshold is reached, which drives up training costs in terms of resources like memory storage or computing time.

To mitigate these very large training costs, approaches based on data representation formats simpler than the Floating Point 32-bit (FP32) standard have been proposed (Courbariaux et al., 2014; Gupta et al., 2015). These approaches successfully mitigate the enormous training costs of DNNs by using data representation formats that either reduce computing costs or diminish the requirements in terms of memory storage and bandwidth. In particular, some of these proposals have shown the benefits of combining half-precision and single-precision compute during training in terms of keeping model accuracy and reducing compute and memory costs (Micikevicius et al., 2017; Kalamkar et al., 2019). These approaches accelerate linear algebra operations by accumulating half-precision input operands to generate 32-bit outputs. While this mixed-precision (MP) arithmetic can successfully reduce the use of resources like memory bandwidth or hardware components like register file size, it has a very limited capacity for diminishing computing costs and it is unable to reduce output data size.

In this paper we propose new training methodologies able to exclusively use half-precision for a large part of the training process, which constitutes a very significant improvement over mixed-precision approaches in terms of compute and memory bandwidth requirements. We propose two different approaches, the first one statically assigns either the Brain Floating Point 16-bit (BF16) or the FP32 format to the model parameters involved in the training process, while the second dynamically switches between BF16 and MP during training depending on its progress. Our approaches

do not require mixed-precision arithmetic while computing linear algebra operations for a large portion of the training process, which enables them to deliver the same performance as if they were operating with half-precision arithmetic during the whole training while providing the same model accuracy as if FP32 was used. This paper is the first in demonstrating that half-precision can be extensively used during DNNs training without the need for mixed-precision arithmetic. We made our code available[1].

## 2    BACKGROUND ON MIXED-PRECISION APPROACHES AND MOTIVATION

Mixed-Precision training has been extensively explored in recent years. Approaches mixing Floating Point 16-bit (FP16) and FP32 datatypes have been proposed (Micikevicius et al., 2017). In these approaches, multiplications of FP16 parameters are accumulated in FP32 registers to minimize data representation range and precision issues. Importantly, relevant phases of the training process like computing weight updates (WU) or dealing with batch normalization (BN) layers entirely use FP32, which implies that a FP32 representation of network weights and biases is kept during the whole training. This approach requires some additional computations to enforce that FP32 values are converted to FP16 without data representation range issues. This approach is used by Nvidia Tesla V100 GPUs via mixed-precision computing units called tensor cores, which are able to multiply FP16 parameters and store the results in FP32. Figure 1a displays the most fundamental operation of this approach combining FP16 and FP32, the mixed-precision Fused Multiply-Add (FMA) instruction, which computes $D = A \cdot B + C$. Input parameters $A$ and $B$ are represented in the FP16 format. The result of the $A \cdot B$ operation is kept in FP32 and added to the $C$ parameter, which is represented in FP32 as well. The final output $D$ is also represented in FP32. FMA instructions constitute around 60% of the whole training workload for several relevant CNN models, as Section 3 shows.

A more recent approach proposes mixed-precision arithmetic combining BF16 and FP32 (Kalamkar et al., 2019). It is very close to its FP16 counterpart with the exception of the full to half precision conversion. Since BF16 has the same data representation range as FP32, conversion from full to half precision is very simple in this case since it just requires applying the Round to Nearest Even (RNE) technique. This approach also processes WU and BN layers with FP32. Figure 1b shows a representation of a mixed-precision FMA combining BF16 and FP32. It is very close to the previously described FP16-FP32 FMA with the only difference being the data representation format of input parameters $A$ and $B$.

While mixed-precision FMA instructions bring significant benefits since they require less memory bandwidth and register storage than FP32 FMAs, there is still a large margin for improvement if an entirely BF16 FMA like the one represented in Figure 1c could be extensively used for training purposes. First, since BF16 FMA requires exactly one half of the register storage of a FP32 FMA, it doubles its Single Instruction Multiple Data (SIMD) vectorization capacity and, therefore, it may significantly increase its FMA instructions per cycle ratio. Extensions of Instruction Set Architectures (ISA) to allow SIMD parallelism are becoming a key element for floating-point performance, which has motivated major hardware vendors to include them in their products (Jeffers et al., 2016; Stephens et al., 2017). Finally, BF16 FMA instructions also bring significant reductions in terms of memory bandwidth since they involve 50% and 25% less data than FP32 and MP FMAs, respectively. While half-precision arithmetic has not been used to train DNNs due to its lack of training convergence, this paper describes two techniques to fully use it while keeping the same convergence properties as FP32. This paper analyzes in detail 3 relevant training workloads in Section 3, and applies these findings to build its two main contributions in Section 4.

## 3    WORKLOAD ANALYSIS

We consider three CNN models: AlexNet, Inception V2, and ResNet-50. Section 5 describes the exact way we use these models and the methodology we follow to analyze their training workload. Figure 2a shows an instruction breakdown of processing one batch on these networks. This figure shows how floating-point instructions constitute a large portion of these workloads. For example, they represent 58.44% of the total in the case of AlexNet. A very large portion of these floating-point instructions, 57.42% of the total, are FMA instructions. In the cases of Inception and ResNet-

---

[1]https://github.com/dynamicprec/dynamic

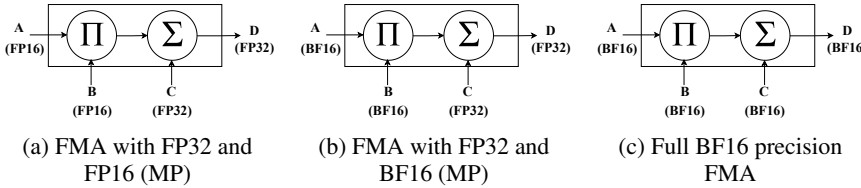

Figure 1: Approaches to compute Fused Multiply-Add (FMA) instructions

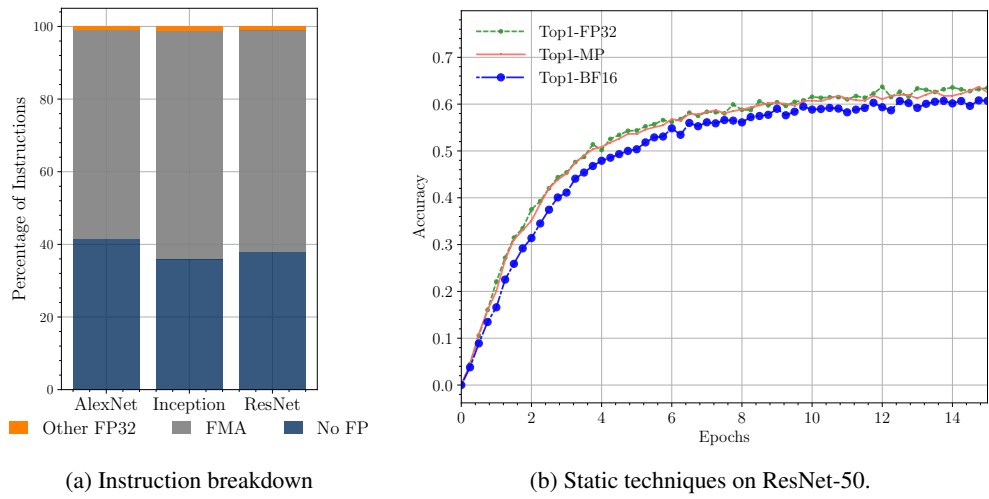

(a) Instruction breakdown

(b) Static techniques on ResNet-50.

Figure 2: CNN Analysis

50, FMA instructions represent 60.93% and 62.95% of the total, respectively. Therefore, FMA instructions constitute a large portion of the whole training workload, while other FP32 instructions represent a small instruction count that remain below 1.1% for these three CNNs. This justifies to focus on FMA instructions, as executing them in half-precision has a large potential for performance improvement.

Prior research (Micikevicius et al., 2017; Kalamkar et al., 2019) describes the need for using 32-bit arithmetic in Weight Updates (WU) and Batch Normalization (BN) layers when using training approaches based on mixed-precision arithmetic. We run an experimental campaign to confirm this observation and to measure the number of instructions devoted to WU and BN. For the case of ResNet-50, this instruction count is around 30 million instructions per batch, that is, just 0.04% of the FP instructions. AlexNet and Inception V2 produce similar results. In conclusion, reducing the cost of FMA instructions has a high potential for very significant performance improvements even if WU and BN layers are computed using full precision arithmetic.

Processing one training batch for the cases of AlexNet, Inception and ResNet-50 requires running 53.3, 37.2, and 70.0 billion dynamic instructions per batch, respectively. The number of model parameters drives the size of these workloads. AlexNet was trained with a batch size of 256, while Inception and ResNet use a batch size of 64.

## 4 PROPOSAL

We propose two training methodologies that rely exclusively on half-precision BF16 for a large portion of the training process, i.e., a large portion of FMA instructions. Prior mixed-precision approaches preclude large gains in computing costs as some of the data elements remain in FP32. However, an FMA entirely relying on BF16 can potentially double the SIMD vectorization through-put of current processors and alleviate memory bandwidth requirements.

We first propose a scheme that performs all FMA instructions in BF16 (see Figure 1c) except those involved in computing WU and processing BN layers, which are entirely performed in FP32. While this method might not deliver the desired level of accuracy for all CNNs, Section 6 shows how it behaves remarkably well for the Inception V2 model, since it obtains the same level of accuracy as state-of-the training using MP and FP32.

However, some CNNs cannot entirely rely on half-precision arithmetic during training. For example, Figure 2b shows *top1* accuracy achieved by three training techniques during 15 epochs for ResNet-50. The first technique (referred as FP32 in Figure 2b) entirely relies in FP32 arithmetic, the second approach (referred as MP in Figure 2b) represents the state-of-the art Mixed-Precision training (Kalamkar et al., 2019), and the third approach (referred as BF16 in Figure 2b) performs all FMA instructions in BF16 except for WU and BN. While the BF16 approach behaves relatively well, it displays lower accuracy than MP and FP32 for all the epochs, which indicates the need for an approach able to take advantage of BF16 arithmetic while delivering the same accuracy results as mixed- or full-precision approaches. The methodology we use to generate Figure 2b is described in Section 5.

Our second contribution dynamically switches between MP and BF16 to deliver the same accuracy as MP while relying in BF16 FMAs during a large portion of the training process. Algorithm 1 displays a high level pseudo-code of our proposal. It starts the training process using the state-of-the art mixed-precision approach (Kalamkar et al., 2019) for several batches, defined by $numBatchesMP$ parameter. Then, it computes the Exponential Moving Average (EMA) (Lawrance & Lewis, 1977) of the training loss and, if its reduction is larger than a certain threshold ($emaThreshold$ parameter), it computes the next $numBatchesBF16$ using BF16 FMAs, except for WU and BN. Once training has gone through these $numBatchesBF16$ batches, our algorithm checks EMA and compares its reduction with the $emaThreshold$ parameter. If this reduction is not large enough, the algorithm switches back to MP arithmetic. Otherwise, it keeps using BF16 arithmetic for $numBatchesBF16$ batches before checking EMA again.

## 5 EXPERIMENTAL METHODOLOGY

### 5.1 EXPERIMENTAL SETUP

Our experiments are performed on Intel Xeon Platinum 8160 processors, which include the AVX512 ISA. We use the Intel-Caffe (Intel, a) framework (version 1.1.6a). We use the Intel MKLDNN (Intel, c) (version 0.18.0) Deep Neural Network library and the Intel MKL library (Intel, b) (version 2019.0.3) to run numerical kernels since both libraries are optimized to perform well on our testing infrastructure. Finally, to define and run the experiments we use the *pyCaffe* python interface, which takes care of loading the data and orchestrating the execution.

### 5.2 EMULATION OF BF16 USING DYNAMIC BINARY INSTRUMENTATION

Due to the lack of available hardware implementing the BF16 numerical format, we rely on an emulation technique to perform our experiments. Several approaches have been used in the past to emulate the behaviour of reduced floating-point representations, most notably via libraries that perform transformations like truncation and rounding (Chatelain et al., 2019; Dawson & Düben, 2017; Kalamkar et al., 2019). We develop a binary analysis tool based on PIN 3.7 (Luk et al., 2005). Our tool captures and instruments dynamic instructions, which enables adaptating numerical operands to the targeted numerical data format. Our approach seamlessly works on complex frameworks like PyTorch, Tensorflow, or Caffe, with interpreted languages, and is able to instrument instructions triggered by dynamically linked libraries. Our binary analysis tool performs the following steps:

- It checks the current operation mode, which can be FP32, MP, or BF16 (see Figure 1).

- It checks the current execution routine to determine if we are executing routines that belong to WU or BN layers. If that is the case, computation proceeds with FP32.

- The tool intercepts the dynamic instructions of the workload and detects all floating-point operations, including FMAs. For each FMA instruction, operands that need to be rounded to BF16, depending on the current operation mode, are rounded using the RNE algorithm.

---

**Algorithm 1** Dynamic Precision

---

1: $numBatchesMP \leftarrow 10$             ▷ Number of consecutive MP batches
2: $numBatchesBF16 \leftarrow 1000$         ▷ Number of consecutive BF16 batches
3: $emaThreshold \leftarrow 0.04$           ▷ Defines EMA reduction threshold
4:
5: $precisionModeBF16 \leftarrow False$     ▷ Indicates current precision mode, $True$ means *BF16*
6: $countBatchesBF16 \leftarrow 0$       ▷ Counts how many *numBatchesBF16* have been executed
7: $numBatchesTrain \leftarrow numBatchesMP$     ▷ Number of batches per training loop iteration
8:
9: **for** $i = 0$ to $niter$ **do**       ▷ Training Loop: *niter* value depends on the number of epochs
10:     *train.step(numBatchesTrain)*    ▷ Execute *numBatchesTrain* batches in *precisionModeBF16*
11:     $trainingLoss[i] \leftarrow train.trainingLoss$
12:     **if** $(i = 5)$ **then**           ▷ Initial history to calculate *EMA*
13:        $EMA \leftarrow average(trainingLoss)$
14:     **if** $(i > 5)$ **then**
15:        $EMAprev \leftarrow EMA$
16:        $EMA \leftarrow emaCalculation(trainingLoss, EMAprev)$     ▷ Each *numBatchesMP*
17:        **if** $(precisionModeBF16! = True)$ **then**
18:           **if** $((EMAprev - EMA) > emaThreshold)$ **then**    ▷ If training loss goes down
19:             $precisionModeBF16 \leftarrow True$
20:             *changeToBF16()*           ▷ Switch precision to BF16
21:        **else**
22:           $countBatchesBF16 \leftarrow countBatchesBF16 + numBatchesTrain$
23:           **if** $(countBatchesBF16 = numBatchesBF16)$ **then**
24:             **if** $((EMAprev - EMA) > emaThreshold)$ **then** ▷ If training loss goes down
25:               $countBatchesBF16 \leftarrow 0$        ▷ Stay in BF16 precision
26:             **else**               ▷ If training loss stagnates
27:               $precisionModeBF16 \leftarrow False$
28:               *changeToMP()*          ▷ Switch precision to MP
29:               $countBatchesBF16 \leftarrow 0$

---

- The tool can dynamically change its operation mode anytime via a simple inter-process communication method that can be invoked from the python high-level interface.

To mitigate the overhead of our binary analysis tool, we implement two optimizations: First, we vectorize the truncation and rounding routines via AVX512 instructions. Second, we avoid redundant rounding and truncation operations by identifying instructions belonging to the same basic block sharing some input operands already stored in the register file. These two optimizations reduce the overhead of the tool from $100\times$ to $25\times$ with respect to native runs of the binary on real hardware.

## 5.3 DYNAMIC AND STATIC TECHNIQUES

This paper considers two different types of training techniques: **static schemes** and **dynamic schemes**. When using static schemes, the training procedure uses the same data representation form for a given parameter during its complete execution. For example, the three techniques displayed in Figure 2b are static. We define the following schemes:

- **MP:** FMA instructions belonging to WU and BN layers always use FP32 precision. The remaining FMA instructions use the mixed-precision approach represented in Figure 1b). This scheme replicates prior work on mixed-precision(Kalamkar et al., 2019).
- **BF16:** FMA instructions belonging to WU and BN layers always use FP32 precision. The remaining FMA instructions use BF16 operands to multiply and to accumulate (Figure 1c).

The *BF16* method is the first contribution of this paper. It extensively uses half-precision arithmetic while displaying good convergence properties.

The *Dynamic* scheme we propose in this paper switches between the *MP* and *BF16* static techniques during training, as explained in Section 4 and detailed in Algorithm 1. This dynamic method im-

proves the training convergence properties of *BF16* while still relying in half-precision FMAs for a very large portion of the execution.

The EMA threshold (*emaTreshold*) is set at 4%. This value is computed as the average EMA reduction when using FP32 computations. The minimum number of batches to be performed in *BF16*, defined by the $numBatchesBF16$ parameter is set to 1,000, which precludes frequent unnecessary transitions between the two schemes. We set the $numBatchesMP$ parameter to 10, which keeps the number of batches using the *MP* regime low while keeping its benefits in terms of convergence.

## 5.4 Convolutional Neural Network Models

To evaluate our proposals we consider the AlexNet (Krizhevsky et al., 2012), Inception V2 (Szegedy et al., 2015) and ResNet50 (He et al., 2015b) models. They are representative CNN state-of-the-art.

We use the ImageNet database (Deng et al., 2009) as training input. To keep execution times manageable when using our binary instrumentation tool, we run the experiments using a reduced ImageNet Database, similar to the Tiny ImageNet Visual Recognition challenge data set (Fei-Fei). Therefore, we use 256,000 images divided into 200 categories for training, and 10,000 images for validation. The images have no modifications in terms of the size. All the evaluated CNN models remain unmodified, the only change is loading a reduced dataset.

AlexNet is selected due to its simplicity in terms of structure and amount of required computations. To train AlexNet we consider a batch size of 256 and the base learning rate is 0.01, which is adjusted every 20 epochs taking into account a weight decay of 0.0005 and a momentum of 0.9. This model is trained for 32 epochs.

We use Inception because it is a model conceived to reduce computational costs via cheap 1x1 convolutions. To train it we use a batch size of 64 and a base learning rate of 0.045, which is updated every 427 steps (0.11 epochs). The gamma, momentum and weight decay are set to 0.96, 0.9, and 0.0002, respectively. The training process is executed for 16 epochs.

Finally we use ResNet-50. It is a network that delivers good accuracy and avoids the vanishing gradients issue by using residual blocks and the MSRA initializer (He et al., 2015a). We train it using a multi-step approach. The batch size is 64 and the base learning rate is 0.05, which is updated every 30 epochs. The gamma hyperparameter, momentum value, and weight decay are set to 0.1, 0.9, and 0.0001, respectively. The training process runs for a total of 32 epochs.

## 6 Evaluation

Figure 3 and Table 1 show results from our evaluation campaign. The x-axis of the three plots belonging to Figure 3 represent the epochs of the training process while the y-axis represents the accuracy reached by the model over the validation set. Table 1 shows the test accuracy we reach for the three network models when using the *FP32* and *MP* baselines and our two contributions: *BF16* and *Dynamic*.

The AlexNet model, due to its structure, shows a good response when lower precision numerical data types are used. As can be seen in Figure 3a all techniques converge, although the *BF16* approach shows the worse accuracy when compared to the *Dynamic* or the *MP* techniques. Table 1 shows that *FP32*, *MP*, *Dynamic*, and *BF16* reach top-5 accuracies of 84.50%, 84.43%, 84.02% and 82.56% for AlexNet after 32 epochs. Importantly, *Dynamic* reaches the same accuracy as *FP32* and *MP* while using the *BF16* approach for 94.60% of the FMAs. In contrast, the *BF16* static technique does

Table 1: Accuracy and percentage of FMAs executed in BF16 precision for evaluated CNNs

| Model | Epoch | FP32 | | MP | | Dynamic | | | BF16 | | |
|---|---|---|---|---|---|---|---|---|---|---|---|
| | | Top-1 | Top-5 | Top-1 | Top-5 | Top-1 | Top-5 | BF16FMA | Top-1 | Top-5 | BF16FMA |
| AlexNet | 32 | 60.79% | 84.50% | 60.18% | 84.43% | 60.32% | 84.02% | 94.60% | 57.80% | 82.56% | 99.93% |
| Inception | 16 | 74.01% | 92.36% | 73.73% | 92.67% | 72.80% | 92.02% | 95.55% | 72.03% | 92.05% | 99.90% |
| ResNet-50 | 32 | 75.97% | 93.37% | 75.76% | 93.20% | 74.20% | 92.70% | 96.40% | 73.00% | 92.30% | 99.91% |

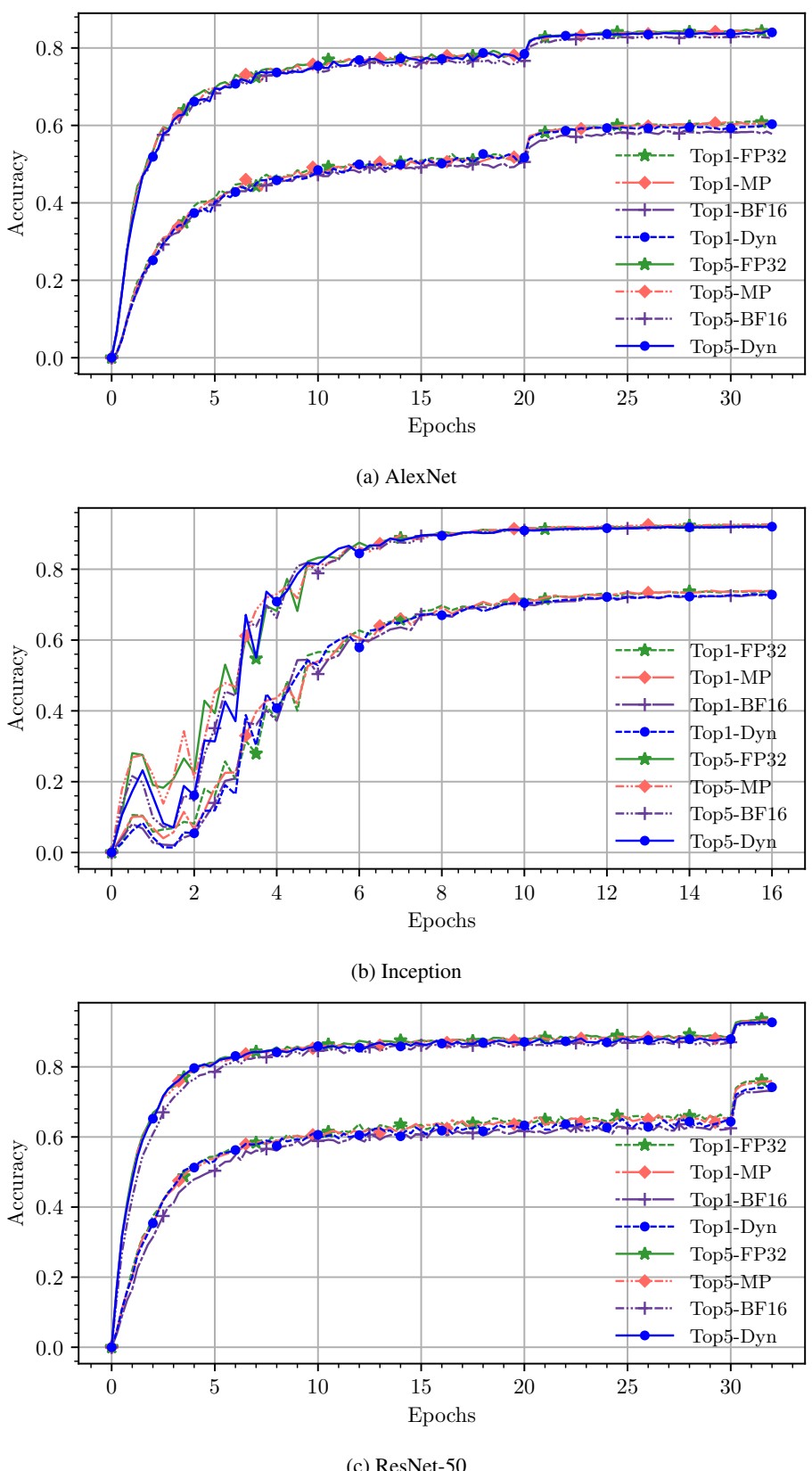

(a) AlexNet

(b) Inception

(c) ResNet-50

Figure 3: Test accuracy achieved by the different training approaches.

99.93% of the FMAs in full BF16 precision (0.07% are in WU and BN layers), but the accuracy drops by almost 3% in Top-1 and 2% in Top-5. This drop in accuracy happens just by doing an additional 5% *BF16* FMAs. This give us some space to improve the *Dynamic* approach by reducing the percentage of *BF16* FMAs with the objective to increase the accuracy of the model.

Figure 3b shows the validation accuracy during 16 epochs for the Inception V2 model. It shows fluctuation on the accuracy evaluation during training due to its structure and hyperparameters tuning. *Dynamic* responds in a robust way to these changes, which highlights its general applicability. Table 1 shows that *FP32*, *MP*, *Dynamic*, and *BF16* reach top-5 accuracies of 93.36%, 92.67%, 92.02%, and 92.05% for Inception V2 after 16 epochs.

Finally, the evaluation on ResNet-50 demonstrates that the *Dynamic* approach is effective when applied to deeper CNNs. In this case, the precision of the model reaches state-of-the-art levels while using half-precision for 96.4% of the FMA instructions. Figure 3c and Table 1 display the exact accuracy numbers we get from our evaluation after 32 epochs. In this experiment the Top-1 accuracy drops just 1.2% comparing the *BF16* and *dynamic* approaches, however we could improve the *dynamic* technique relaxing the quantity of *BF16* FMA executed to gain more accuracy.

## 6.1    SENSITIVITY ANALYSIS FOR DYNAMIC PRECISION ALGORITHM

We provide a sensitivity analysis for the parameters employed in Algorithm 1. The objective is to show that for a range of reasonable parameters the algorithm behaves as expected. To do this analysis we set one of the parameters to the currently used value (*numBatchesMP* to 10) to have a manageable number of combinations. We then test all the possible combinations using *numBatchesBF16* = {500, 1000, 2000} and *emaThreshold* = {0.02, 0.04, 0.08}, that is, a total of 9 different combinations. As stated in Section 5.3, during our evaluation we used the configuration {*numBatchesMP*, *numBatchesBF16*, *emaThreshold*} = {10, 1000, 0.04} for all the evaluated networks.

Figure 4 shows, for a number of *ResNet-50* epochs, the accuracy obtained for each of the 9 tested configurations as part of the sensitivity analysis. The name convention for these configurations is *Dyn-<emaThreshold>_<numBatchesBF16>*. In addition, we include accuracy for BF16, MP, and FP32 executions.

As shown in the figure, the accuracies obtained at each epoch are always above that of the BF16 technique. For early epochs (i.e., 2 and 4) the dynamic configurations remain between BF16 and FP32 accuracy, or even slightly above FP32, due to initial noise. As training advances all dynamic techniques behave similarly and present accuracies that are above BF16 and similar to those obtained with MP and FP32, as we would expect. The most important parameter is the *emaThreshold*, as it decides when a precision change occurs. As long as this parameter is reasonably set to detect training loss improvement or degradation the algorithm is bound to behave as expected.

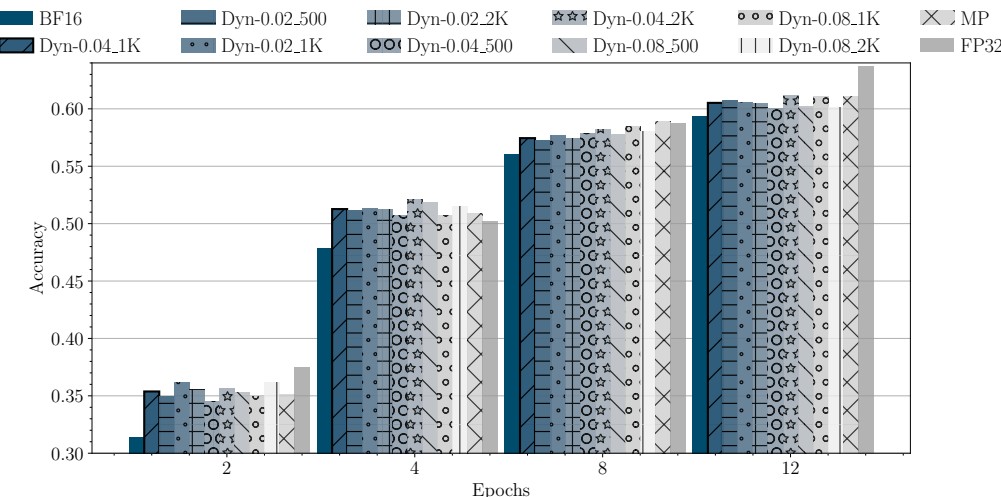

Figure 4: Sensitivity analysis to dynamic precision algorithm parameters on ResNet-50.

## 7 RELATED WORK

Prior work indicates that dynamic fixed-point is effective to train deep neural networks with low precision multipliers (Courbariaux et al., 2014). This approach obtains state-of-the-art results by uniformly applying the dynamic fixed point format with different scaling factors, which are driven by the overflow rate displayed by the fixed-point numbers. Our proposals target deeper neural networks than this approach and do not uniformly apply the same format to all network parameters. Instead, we differentiate between computations requiring FP32 during the whole training, like weight updates, from the ones that are well-suited for dynamic data representation schemes.

Previous approaches show the benefits of applying stochastic rounding to 16-bit fixed-point multiply and add operators Gupta et al. (2015). This previous work rely on FPGA emulation to show the benefits of stochastic rounding when applied to a custom fully connected neural network dealing with the MNIST dataset. The authors also consider a CNN similar to LeNet-5 to enlarge their experimental campaign.

Previous approaches propose a training process of DNN using 8-bit floating point numbers (Wang et al., 2018). They rely on a combination of 8-bit and 16-bit and additionally using stochastic rounding to obtain state-of-the-art results. The neural networks used in this previous approach are much simpler than the ones we consider in this paper, which do not allow 8-bit arithmetic.

The BF16 numerical format has been applied to specific-purpose hardware targeting deep neural networks (Wang & Kanwar, 2019). This specific hardware used an approach very similar to the mixed-precision techniques described in the paper by Kalamkar et al. (2019) and, consequently, our *Dynamic* approach can be applied on top of them to reduce computing costs.

## 8 CONCLUSIONS AND FUTURE WORK

This paper analyzes the instruction breakdown of workloads focused on deep neural network training that rely on mixed-precision training. We show that mixed-precision FMAs constitute around 60% of these workloads and propose two approaches based on half-precision FMAs to accelerate the training process without hurting accuracy.

The first approach uses BF16 FMAs for most of the training workload, except routines involved in weight updates or batch normalization layers. This approach uses BF16 for more than 99% of the FMAs, which has a very strong potential for performance improvement, while reaching slightly smaller accuracy than the state-of-the-art. We propose a second approach that dynamically switches between different data representation formats. This dynamic approach uses BF16 for around 96% of the FMAs while reaching the same precision levels as the standard single-precision and mixed-precision approaches.

Our two proposals are evaluated considering three state-of-the-art deep neural networks and a binary analysis tool that applies the required precision for each instruction. To the best of our knowledge, this is the first paper that demonstrates that half-precision can be used extensively on $\geq$94% of all FMAs during the training of very deep models without the need for mixed-precision arithmetic.

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
