# OpenReview forum: "A Dynamic Approach to Accelerate Deep Learning Training"
_ICLR.cc/2020/Conference — Reject_

### Official Review · AnonReviewer2 · 2019-10-22
**Official Blind Review #2**

**Rating:** 3

**Review:**

The author(s) propose to accelerate the training of deep neural networks while also maintain the performance of the trained model by switching between fully half-precision computation and mixed-precision computation. Compared to the commonly-used mixed-precision training strategy, the proposed method can accelerate the training speed. Besides, on an image classification task, models trained by the proposed method achieve comparable performance with those trained by mix-precision training or full-precision training.

Strength:
1.	Section 3 provides useful information for the workloads of deep neural networks.

Weakness:
1.	The overall idea is not novel. The proposed method simply switches between two existing training strategies, i.e., the mixed-precision training and half-precision training. The claim that "this paper is the first in demonstrating that  half-precision can be used for a very large portion of DNNs training and still reach state-of-the-art accuracy" may not correct, in fact, Nvidia's apex has already supported using mixed-precision or entirely half-precision to train DNNs, and there is no clear evidence that the proposed method is better than theirs due to the lack of experiments on more tasks and datasets.


2.	From Table 1, the Dynamic strategy outperforms BF16 in terms of classification accuracy. However, from the experiments, it’s unable to tell that the gains actually come from this Dynamic strategy. Maybe similar gains can be obtained once the same amount of MP iterations are executed at any period of the training process. For example, consider a simpler strategy: first train the model with BF16 for K% of the total training iterations, then for the last (100-K)% iterations, train it with MP. K can be tuned so that the proportion of BF16FMA is close to those in Table 1. Models trained with this strategy might achieve similar performance with the proposed Dynamic strategy.


3.	It’s hard to apply the proposed method in real applications, since BF16 is only supported by very few kinds of hardware.


4.	To demonstrate the effectiveness of the proposed method more clearly, it could be better to provide the exact proportion of reductions in terms of memory/computation/bandwidth in the experiments.


5.	From Table 1, there still exists a large performance gap in terms of accuracy (1.56% for ResNet-50) between the model trained by the proposed method and the model trained by state-of-the-art MP.


6.	The organization of this paper can be improved. For example, Section 5.2 spends too much space introducing the emulation of BF16, which I think is not very relevant to the topic of this paper. And Figure 3 takes too much space.


**Experience Assessment:**

I have read many papers in this area.

**Review Assessment: Checking Correctness Of Derivations And Theory:**

I carefully checked the derivations and theory.

**Review Assessment: Checking Correctness Of Experiments:**

I carefully checked the experiments.

**Review Assessment: Thoroughness In Paper Reading:**

I read the paper at least twice and used my best judgement in assessing the paper.

---

> ### Author Response · Authors · 2019-11-12
> **Answers to blind review #2**
>
> First just to say thanks to you for the feedback you are giving us.
>
> 1. Our approach is the first one to use BF16 precision for a large part of the training and keep the same accuracy as FP32. It dynamically identifies when to switch to MP to avoid accuracy reductions.
> Nvidia does support FP16 precision; however, as Nvidia’s apex API documentation (https://nvidia.github.io/apex/amp.html) states it “may not achieve the stability of the true mixed precision options”. Our approach dynamically switches between half and mixed precision to obtain the comparable accuracy results as state-of-the-art techniques while using the cheap BF16 arithmetic for most of the training.
>
> 2. The advantage of our dynamic approach is to identify when BF16 precision is not sufficient to lower training loss further, for example due to a local minimum. At that point, switching to a higher precision method can help overcome that local minimum and later on we can switch back to BF16 precision which would be enough to lower the training loss function again.
> To prove this point, we have performed the suggested test. We train AlexNet during the first 94% epochs, which corresponds to the first 30 epochs, using the BF16 approach. From that point, we train the Network using the MP approach. At epoch 32 we obtain a top1 accuracy of 58.89%, which is smaller than the 60.32% accuracy we obtain with our technique. This experiment demonstrates that enabling our dynamic approach during training enables significant execution time reductions while keeping the same accuracy as state-of-the-art.
>
> 3. The BF16 data type is being adopted gradually by most major DNN frameworks. Regarding hardware support, it is going to be widely available in Intel Xeon-class machines in a few months (e.g., Cooper Lake first half 2020), and also added in the next update of the ARMv8-a ISA definition. Therefore, our proposed method will soon be applicable to a wide range of machines. We believe that research to find novel training techniques that can ease training process execution time is of interest to the community.
>
> 4. We need to rely on dynamic binary instrumentation to be able to faithfully emulate the BF16 data type and the round to nearest truncation. The overheads imposed by this instrumentation preclude us from gathering these measurements.
> We agree that measurements would be of interest. However, by providing the number of instructions executed with each data type and considering that the vectorization throughput is doubled using BF16 when compared to MP, we aim to show that there is a clear reduction in terms of memory, compute, and memory bandwidth.
>
> 5. We agree that reaching state-of-the-art accuracy is important, in AlexNet we do reach state-of-the-art accuracy with the same number of epochs. As pointed out, in ResNet50 with the same number of epochs there is a gap of 1.56% (75.76% MP vs 74.20% Dynamic). However, one can train faster with Dynamic, and if more accuracy is desired, run a few more epochs.
> We have run a few additional epochs using our Dynamic approach and reached the following differences between MP and Dynamic top-1 accuracies at the end of each epoch. For epochs {32..40} = {1.56, 1.61, 1.64, 1.17, 1.12, 1.19, 1.28, 1.67, 1.10}. We can trade-off more accuracy by running a few additional epochs using a faster training process.
>
> 6. We agree. We will fix this in the paper.

---

### Official Review · AnonReviewer3 · 2019-10-23
**Official Blind Review #3**

**Rating:** 3

**Review:**

In this paper, the authors propose approaches to accelerate deep neural network training with mixed-precision arithmetic.
Observed that relying purely on half-precision arithmetic results in lower accuracy, the authors developed a method
to dynamically switch between mixed-precision arithmetic (MP) and half-precision arithmetic (BF16 FMA).
Empirical results show that the dynamic approach can achieve similar accuracy as MP and FP32 algorithms.

Although this paper shows the possibility to accelerate DNN training without great loss in performance, there are many issues with the paper itself. First, the title is ambiguous. The dynamic approach could mean a lot of things while training
DNNs and one cannot tell what the paper is about simply relying on the title.
Also, the dynamic algorithm itself is not well presented. For example, how to choose the hyperparameters? What is the
overhead to switch between MP and BF16FMA?

Apart from the algorithm itself, I also have questions regarding the experimental results.
Is there a reason why the performance of the half-precision arithmetic
varies across different neural networks (Inception > Resnet)?
Specifically, what is the key factor that influences the sensitivity of a neural network towards precision?

Overall I think this paper should be improved in its experiments and presentation.



**Experience Assessment:**

I do not know much about this area.

**Review Assessment: Checking Correctness Of Derivations And Theory:**

I assessed the sensibility of the derivations and theory.

**Review Assessment: Checking Correctness Of Experiments:**

I assessed the sensibility of the experiments.

**Review Assessment: Thoroughness In Paper Reading:**

I read the paper at least twice and used my best judgement in assessing the paper.

---

> ### Author Response · Authors · 2019-11-12
> **Answers to blind review #3**
>
> Thanks for your review, we are taking the comments to improve our research.
>
> --> The title is ambiguous ...
> We are going to change the title of the paper on the PDF file. We were asking to the organizers and they answered us that we could change the title on the generated pdf but unfortunately on the OpenReview platform is not possible.
>
> --> For example, how to choose the hyperparameters?
> We selected the evaluated hyperparameters by analysing FP32 executions. We checked the evolution of the training loss to determine how it improves between batches in order to select sensible hyperparameters. As long as the emaThreshold parameter is reasonably set to detect training loss improvement or degradation the algorithm will behave as expected. We want to emphasize the fact that the parameters we use in the paper are the same for the three evaluated networks, which reinforces our believe that these parameters are not sensitive to different networks.
> In addition, we are performing a sensitivity analysis for these hyperparameters and will include it in the paper. The analysis will focus on ResNet50. We set one of the parameters to the currently used value (numBatchesMP to 10) to have a manageable number of combinations. We then test all the possible combinations using numBatchesBF16 = {500, 1000, 2000} and emaThreshold = {0.02, 0.04, 0.08}.
>
> --> What is the overhead to switch between MP and BF16FMA?
> On a real system that supports these modes the overhead is negligible. For example, one would need to write to a hardware register to indicate the current mode, which is an operation that takes a few cycles, compared to millions of floating-point operations between each switch.
>
> --> … varies across different neural networks (Inception > Resnet)?
> Inception tolerates low-precision arithmetic better than ResNet50 since it extensive uses batch normalization layers, which mitigate numerical precision issues. Observing a better behaviour of Inception than ResNet when half-precision arithmetic is used is expected.
>
> --> … sensitivity of a neural network towards precision?
> Multiple factors may influence network sensitivity to low-precision. As we mentioned above, certain network components like batch normalization layers may successfully mitigate numerical precision issues. Other approaches, like dropout, may also contribute in reducing network tolerance to low precision scenarios.

---

### Official Review · AnonReviewer1 · 2019-10-24
**Official Blind Review #1**

**Rating:** 3

**Review:**

This proposes two techniques to replace mixed-precision arithmetic with half-precision training for a large part of the training process. In the first approach, the authors simply switch all mixed-precision operations with half-precision operations, and can achieve performances slightly lower than SOTA. In the second approach, the authors propose to dynamically switch between mixed-operations and half-precision operations during training. The authors claim that this second approach can match SOTA results while using half-precision arithmetic for more than 94% of training.

Overall the paper is fairly well written, and easy to follow. The proposed techniques seem to empirically work well. However, I have a number of concerns about the paper, which explains my score. I list these concerns below.

1. The proposed approach has a number of additional hyperparameters, which makes it less likely for the algorithm to be widely used if the algorithm is very sensitive to the values of this. For very extreme values of these hyperparameters, I would expect the algorithm to start behaving quite poorly. But it would help a lot to provide some sensitivity analysis to these hyperparameters for reasonable values of these hyperparameters.

2. How much do the optimal hyperparameters (like numBatchesMP, numBatchesBF16, emaT) vary across problems?

3. How much do the above-mentioned optimal hyperparameters vary with mini batch size?

4. How are the other hyperparameters like learning rates selected? Is the learning rate tuned?

5. Are the experiments repeated multiple times?

6. It seems a bit weird to call a modification that simply uses half-precision arithmetic for most FMA operations a significant contribution of the paper, especially since it can't reach SOTA performance.

7. Algorithm 1 should be written out in a better way that shows the training loop. It is slightly confusing the way it is written up right now.

Overall I think the paper would significantly benefit from a more thorough empirical evaluation.

=============================

Edit after rebuttal:
I thank the reviewers for their response and for running the additional experiments. However, I find the updated version of the paper to be inadequate in fully answering my concerns. While the authors have included a hyperparameter sensititivity analysis, I find the experiment to be unconvincing. Only two of the three hyperparameters are swept over a very small range of values, and the results presented are only for the first 12 epochs, while the actual model is typically trained for 90 epochs. While I appreciate the added experiment and realize that 10 days is too short a time to put in a proper sensitivity analysis, based on the current draft of the paper, I cannot recommend accepting this paper. I am however raising my score to a weak reject.


**Experience Assessment:**

I have published one or two papers in this area.

**Review Assessment: Checking Correctness Of Derivations And Theory:**

I assessed the sensibility of the derivations and theory.

**Review Assessment: Checking Correctness Of Experiments:**

I carefully checked the experiments.

**Review Assessment: Thoroughness In Paper Reading:**

I read the paper at least twice and used my best judgement in assessing the paper.

---

> ### Author Response · Authors · 2019-11-12
> **Answers to blind review #1**
>
> Thanks for the feedback and to take the time to do the review of this paper.
>
> 1. We are performing a sensitivity analysis for these hyperparameters and will include it in the revised version of the paper. The analysis will focus on ResNet50. We set one of the parameters to the currently used value (numBatchesMP to 10) to have a manageable number of combinations. We then test all the possible combinations using numBatchesBF16 = {500, 1000, 2000} and emaThreshold = {0.02, 0.04, 0.08}, that is, a total of 8 different combinations. For the whole paper we use the configuration {numBatchesMP, numBatchesBF16, emaThreshold} = {10, 1000, 0.04} for all the considered networks.
>
> The most important parameter is the emaThreshold, as it decides when a precision change occurs. As long as this parameter is reasonably set to detect training loss improvement or degradation the algorithm will behave as expected. We want to emphasize that we use the exact same values for parameters numBatchesMP, numBatchesBF16, and emaThreshold for the three evaluated networks, which highlights the fact that these parameters are network independent.
>
> 2. As stated above, we use the same hyperparameter values to train AlexNet, Inception and ResNet50: numBatchesMP = 10, numBatchesBF16 = 1000, emaThreshold = 0.04.
>
> 3. For all the CNN models evaluated we use the default hyperparameters, including the default mini-batch size, in order to facilitate replication and comparison of results with prior and future studies. If different mini-batch sizes need to be used, one can maintain the behaviour of the dynamic precision algorithm by adjusting the numBatchesMP and numBatchesBF16 parameters. For example, if the mini-batch size is doubled, then by halving the two numBatch parameters the behaviour would be the same. Note that the emaThreshold value would not be changed.
>
> 4. As mentioned before, we use the same hyperparameters as state-of-the-art for each evaluated network. Therefore, learning rates evolve according to these default policies, i.e., ResNet50 changes the rate at epoch 30. We have not tuned the learning rate to our dynamic algorithm.
>
> 5. We use the same random seed to initializate the weights. Using the same random seed lets us to compare results across different scenarios and confirm the quality of our approach.
>
> 6. As stated in the paper, using half-precision arithmetic in all FMA operands can double the computational throughput when using vectorization (e.g., Intel AVX), which is the common case in math libraries like Intel MKL. Approaches like MP preclude efficient code vectorization as one of the operands is 32-bits wide, effectively halving vectorization performance. Therefore, the proposed Dynamic approach (~94% FMAs using BF16) can train networks significantly faster.
>
> We agree that reaching SOTA accuracy is important, in AlexNet we do reach SOTA accuracy with the same number of epochs. In ResNet with the same number of epochs there is a gap (75.76% MP vs 74.20% Dynamic) of 1.54% . However, one can train faster with Dynamic, and if more accuracy is desired, run a few more epochs. For example we have run 8 additional epochs using Dynamic on ResNet50, reaching a difference of 1.10% accuracy test compared with SOTA numbers.
>
> 7. We agree. We will fix the presentation of Algorithm 1 to include the training loop.

---

### Author Response · Authors · 2019-11-15
**Final Revision of the Paper**

We really appreciate the time that all reviewers had with our paper.

We did the changes suggested by the reviewers in the paper PDF:

1. We added the section 6.1 Sensitivity Analysis for Dynamic Precision Algorithm and a new plot with tests related to the behavior of our approach with several values in the new hyper-parameters.

2. We re-organized the Algorithm 1 in order to show the training loop.

3. We did an adjustment to the title of the paper in the final revision.

Thanks for all the feedback.

---

### Decision · Program_Chairs · 2019-12-19

**Decision:**

Reject

**Comment:**

The submission proposes a dynamic approach to training a neural net which switches between half and full-precision operations while maintaining the same classifier accuracy, resulting in a speed up in training time. Empirical results show the value of the approach, and the authors have added additional sensitivity analysis by sweeping over hyperparameters.

The reviewers were concerned about the novelty of the approach as well as the robustness of the claims that accuracy can be maintained even in the accelerated, dynamic regime. After discussion there were still concerns about the sensitivity analysis and the significance of the results.

The recommendation is to reject the paper at this time.